# Assessment of the Durability of Threaded Joints



**Žilvinas Bazaras [1], Mindaugas Leonavičius [2], Vaidas Lukoševičius [1],\* and Laurencas Raslavičius [1]** 

[1] Department of Transport Engineering, Faculty of Mechanical Engineering and Design, Kaunas University of Technology, 51424 Kaunas, Lithuania; zilvinas.bazaras@ktu.lt (Ž.B.); laurencas.raslavicius@ktu.lt (L.R.)

[2] Department of Mechanical and Materials Engineering, Faculty of Mechanics, Vilnius Gediminas Technical University (VILNIUS TECH), 03224 Vilnius, Lithuania; mindaugas.leonavicius@vgtu.lt

\* Correspondence: vaidas.lukosevicius@ktu.lt

**Abstract:** The article deals with the determination of the resistance to cyclic loading of the threaded joints of pressure vessels and defective elements according to the failure mechanics criteria. Theoretical and experimental studies do not provide a sufficient basis for the existing calculation methods for the cyclic strength of the threaded joints of pressure vessels. The short crack kinetics in the threaded joints, a shakedown in one of the joint elements of pressure vessels, i.e., in the bolt or stud, has not been studied sufficiently. The calculation methods designed and improved within the study were based on theoretical and experimental investigations and were simplified for convenient application to engineering practice. The findings could be used to investigate the shakedown of studs of a different cross-section with an initiating and propagating crack. Value: the developed model for the assessment of durability of the threaded joints covers the patterns of resistance to cyclic failure (limit states: crack initiation, propagation, final failure) and shakedown (limit states: progressive shape change and plastic failure). Analysis-based solutions of plastic failure conditions and progressive shape change were accurate (the result was reached using a two-sided approach; the solutions were obtained in view of the parameters of the cyclic failure process in the stud (bolt) and based on experimental investigations of the threaded joints).

**Keywords:** low-cycle fatigue; pressure vessels; progressive shape change; shakedown; statistical assessment; threaded joints

## 1. Introduction

The improvement in the durability and reliability of power, chemical, transport, mining equipment, and machinery structures is related to a variety of factors that influence the limits and determine the strength of individual elements or the entire structure. Failures in such objects lead to the suspension of equipment operations, increasing the risk of accidents, and causing considerable financial damage. Failures that occur are signs of shortcomings in the design, production, and maintenance methods used for a large number of objects. Determining or reviewing the service life of potentially hazardous objects designed or already in service requires the assurance of the strength of each element by making full use of the properties of the material [1–5].

The shell-shaped structural elements of pressure vessels and other fitted large parts (lids, stands, supports) are subjected to various loads. The stress–strain state that develops in an environment characterized by workmanship and operational defects can considerably affect the strength of the elements. Most of the workmanship defects (pores, cracks, and inclusions) appear in the casting process. Operational defects (primarily fatigue cracks) appear under normal and disturbed operating conditions [6,7]. The successful interruption of fatigue cracks and the limit state of the process (that is, the appropriate conditions for crack propagation to stop) have a considerable effect on the safe operation of the structures.

If a structure is free of defects or defects have occurred during operation, the defects shall not reach the critical values to ensure safe operation of the structure. Evaluation of the degree of defects in threaded joints is a complex process that requires both theoretical and experimental investigation after the identification of the defect or crack [8–10].

An increase in the dimensions of the threaded joint may result in not only greater dimensions of the assemblies but also considerable changes in the force flows within the assembly as a result of which the critical internal forces may increase instead of decrease. The degree of loading on the connecting elements of the pressure vessels and the effect of adjacent parts of the actual structure can lead to plastic deformation in the threaded joints. In the event of cyclic variation thereof, the parameters of the non-elastic strain process limit the service life of the structure. If the residual strains resulting from plastic flow at the beginning of loading no longer accumulate after a certain number of cycles, this means that the structure has adapted to the loads [11–13].

The research articles are mostly focused on axial tensile fatigue questions [14–16]. Jiao et al. [17], to investigate the fatigue performance of M24 grade 8.8 high-strength bolts in constant amplitude, tested 21 high-strength bolts in axial tensile fatigue at a stress ratio of 0.5. Based on the experimental results, S-N curves were obtained by regression analysis, and a fractographic analysis was performed on representative specimens. The results were compared with existing experimental results and standards for fatigue. It was concluded that the area of the fatigue propagation region is essentially inversely proportional to the stress ratio. Maljaars et al. [18] presented a meta-study on bolts and bolted connections where a few thousand fatigue tests on these elements were evaluated. The results of this study have been implemented in the new revision of the European standard EN 1993-1-9. This paper provides the background for the modifications. Yu et al. [19] mainly studied the low-cycle fatigue life of the pre-tightened bolts working at a high temperature. A novel test fixture was designed for fatigue tests, and low-cycle fatigue tests of pre-tightened bolts were conducted at the temperatures of 550 °C and 650 °C. The research results provided a theoretical basis for the low-cycle fatigue life prediction of pre-tightened bolts.

The article published by Noda et al. [20] is insightful as it deals with the issues of fatigue strength improvement and the anti-loosening performance of JIS M16 bolt–nut connections. Three different root radii were considered coupled with three different pitch differences. The extension of the root of the radius of the bolt root improved the fatigue limit of the bolt by more than 30% because both the stress amplitude and the mean stress can be reduced. Other articles deal with anti-loosening problems [21–25] or analyze the performance of specific threaded joint fatigue strength [26–29].

Some research focuses on the application of finite element modelling to the study of threaded joints. Okorn et al. [30] provided analytical calculations according to Verein Deutscher Ingenieur (VDI) Recommendation 2230 and numerical analysis using the finite element method (FEM). To verify the FEM analysis, the forces in the bolts were measured under preload and under service load on a test bench. On the basis of analytical and numerical results, the influence of the point of application of the working load on the bolt load and its fatigue life were analyzed for different cases.

In [31] a finite element model was proposed for the variation of the wear profile of a threaded surface to simulate the self-loosening of the bolted joints under transverse loading. The finite element model successfully simulated the phenomenon by which the clamping force gradually decreases due to the loosening of the wear as the distribution and magnitude of the contact stresses between the threads change. The predicted results were in good agreement with the experimental results. The paper [32] presented the finite element analysis model applying the elastic–plastic finite element analysis for bolts under variable combustion pressure. The stress history was then used to calculate the stress intensity factors and the fatigue life of the bolts. The numerical results show that the existence of a crack at a depth of 0.35 mm is the source of premature fracture failure. Article [33] presents the self-loosening mechanism of the bolt in the curvic coupling due to structural ratcheting under cyclic load. Finite element simulations were performed

under several loading conditions with different preloads and cyclic torque loadings in the curvic coupling.

Calculation methods for the cyclic strength of threaded joints in critical objects have not been sufficiently substantiated using theoretical and experimental investigations, or investigations have not been conducted at all. Plain, volumetric, and calculated model investigations using photoelasticity, the finite element method, and other methods do not reproduce the actual operating conditions of the threaded joints. The kinetics of short cracks in threaded joints has virtually not been investigated. The shakedown in one of the joint elements (bolt or stud) has not been investigated. Theoretical and experimental investigations in the area of resistance to cyclic failure and shakedown are insufficient to support the calculation methods and guarantee the strength of the structural elements. Therefore, improving cyclic strength and shakedown assessment methods is important to ensure the greater reliability and better occupational safety of objects designed and already in service.

The paper followed the objective of determining the patterns of resistance of heavily loaded threaded joints and elements of pressure to low-cycle and multi-cycle failure and developing the improved, well-substantiated cyclic strength and shakedown calculation methods. The aim of the paper is: (i) to develop an original methodology for an experimental investigation of threaded joints; (ii) to experimentally explore the stress state of defective threaded joints and elements and the crack initiation conditions in the concentrator environment; to compare the experimentally investigated threaded joints to the normative calculations; to predict the patterns of crack initiation and propagation in threaded joints; (iii) to analytically and experimentally analyze the shakedown of threaded joints in view of the cyclic failure process parameters.

## 2. Research Object

The practice of operation of heavy loaded power, chemical, transport, and mining equipment and machinery, structures shows that fatigue failure of threaded joints between the elements (the lid and body) is often caused by bending loads that are not always predicted or that have not been appropriately assessed in terms of their effect. The bending of the studs or bolts in the flanged and fastening joints occurs due to structural specifics or direction of load. Here, the nominal stress of studs reaches or exceeds 380–450 MPa, and the number of cycles to crack initiation or final failure is within the area of low-cycle failure. The joints are subjected to up to $10^3$–$5 \times 10^3$ load cycles during tightening and to overloads in cases of hydraulic testing, start-up, shutdown, full or partial cooling, change of efficiency and capacity, and in certain emergency and servicing of the equipment. Additional axial cyclic loading appears as a result of initialization, shutdown, change in operating mode, and lid strain. The latter also causes the bending moment. Depending on the function of the bolt and the stud joint, the tightening may reach $0.65\sigma_y$ in various units. The structure of critical threaded joints used in mineral grinding mills is shown in Figure 1.

During drum rotation (diameter: more than 10 m; rotational speed: up to 10 rpm), the bolted and stud joints are subjected to cyclic stress and some are subjected to bending. These joints must be protected against environmental impact (production dust penetrating into the threads).

Workmanship defects (casting pores, pits, and cracks) are unavoidable due to the geometrical form of the body parts, lids, stands, supports, and other elements of the mill. Operational strain gauges show that during mill drum rotation, the stress variation in the retaining surfaces occurs due to the negative asymmetry cycle (cycle asymmetry coefficient $r_a = \sigma_{min}/\sigma_{max} = 0.6 - 0.7$). The variable tension strain state that develops in the environment of workmanship defects requires special investigation. Due to the considerable mechanical and thermal effect, the element of the structure may show low-cycle fatigue cracks, and residual strain may accumulate. In this case, the parameters of the variable nonelastic strain process limit the designed durability. If the residual strain

resulting from the plastic flow at the start of loading no longer accumulates after a certain number of cycles, this means that the structure has adapted to the load.

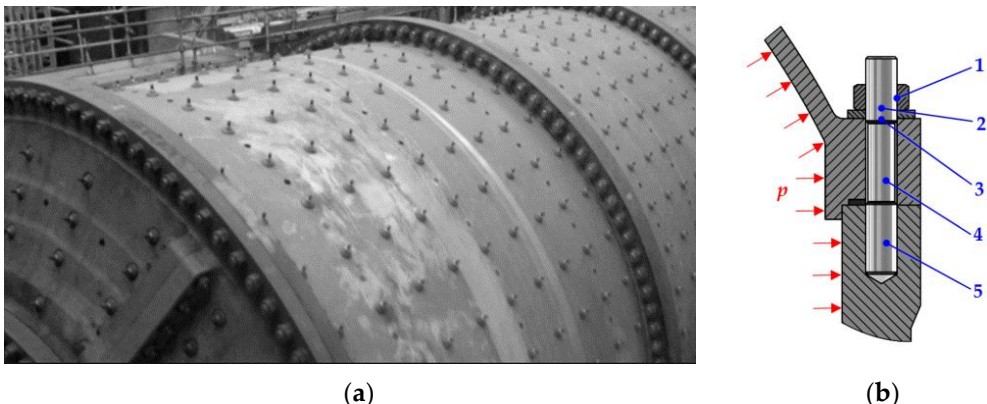

(**a**)          (**b**)

**Figure 1.** Fragment of the mineral grinding mill: (**a**) stress concentration areas in the pressure vessel stud joints; (**b**): 1—in the stud (bolt)–nut joint; 2—in the free-threaded region; 3—in the transitional threaded region; 4—in the smooth region (due to unevenness occurring during surface processing); 5—in the stud-body joint.

The experimental investigation conducted on threaded joints has shown that in certain cases, the total stress caused by the tightening and cyclic bending reaches the plastic state and penetrates to a certain depth. Further repetitive variable loading affects the strength and reliability of the joint. It has also been noticed that the cyclic plastic strain no longer accumulates, i.e., the joint shakedown has taken place as soon as a favorable field of residual stress has developed after a small number of cycles. In the event of the violation of the shakedown conditions, plastic strain of the variable sign (usually local) may occur, or single-sign strain may grow in each cycle (progressive shape change), covering the entire element of the structure or a part thereof. Theoretical articles by the authors on the topic of slack provide a review of the cyclic strength margin of threaded joints.

## 3. Experimental Results and Their Analysis

### 3.1. Experimental Procedures

The experimental investigation was carried out on M52×4 joints made of 38XH3MA steel (chemical and mechanical properties according to the standard GOST 5632-2014: $\sigma_{0.2} = 800$ MPa, $\sigma_u = 910$ MPa, $E = 2.07$ GPa, $\psi = 59\%$, with normalization heat treatment). The stud thread root radius $r_c = 0.4 \div 0.6$ mm.

A special testing methodology and designed equipment were developed intended for thread joint testing, and stiff tension-bending conditions were reproduced.

The especially self-designed fixture with the threaded joint was fitted in the TIRA test machine TIRA test 2300, max. load 100 kN (TIRA GMBH, Schalkau, Germany) and subjected to the axial load (Figure 2).

Furthermore, the following measurement tools were used: dynamometer (in TIRA test 2300)−±0.5% accuracy; discontinuous deformation meter DDA 50 (50 mm base)−±1% accuracy; computer metering system with analog-to-digital converter SPIDER-8; data registration and processing program CATMAN—EXPRESS.

The threaded joint consisting of the stud (1) and two nuts (2) was tightened (Figure 2a). The desired tension value and tightening of the intermediate parts were achieved by periodically tensioning the stud and turning the nut. The nuts (2) rested on the supports (3), with the intermediary parts (4) and rollers (5) between the latter. These enabled the supports (3) to turn in relation to the intermediary parts. The force *F(t)* acting in the direction of tension (Figure 2b) generates a bending moment *M(t).* A force acting in the compression direction produces a bending moment *M\*(t).* The supports (3) and the nuts (2)

were turned over the hinge during the clamp of the upward and downward movement of the clamp of the testing machine, and the stud (1) was bent at the same time.

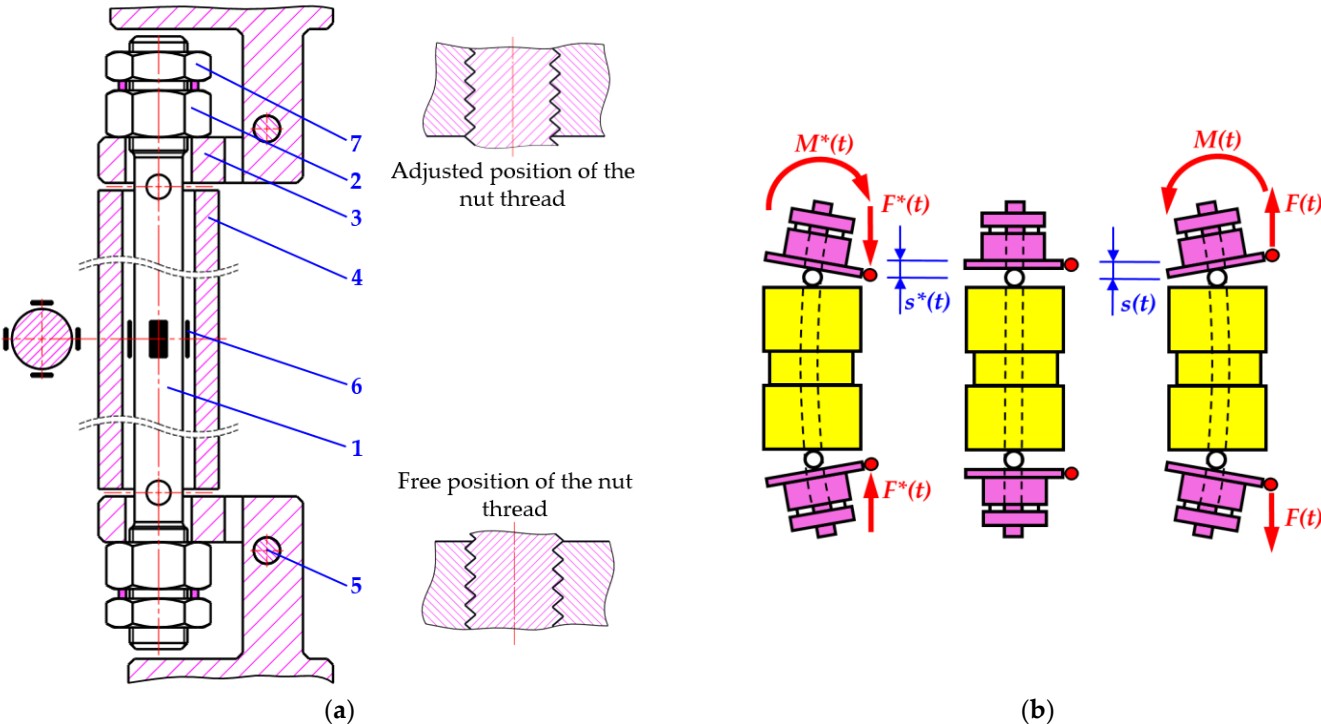

**Figure 2.** Scheme (**a**) and operating principle (**b**) of the special designed bending equipment.

Two strain gauge sensors (6) were attached to the opposite sides of the sample with their axes coincident with the plane of the neutral layer. They were used to measure the tightening stress and its change during the test. The deformation in the outer layers of the stud was measured by two other strain gauge sensors (6), and the extreme stresses $\sigma_{max}$ and $\sigma_{min}$ were determined.

The displacement values $s(t)$ and $s^*(t)$ were measured during the first cycle, when the predicted stresses $\sigma_{max}$ and $\sigma_{min}$ were achieved. The initiation and propagation of the crack in the stud was determined by the fluorescent magnetic particles method (ISO 9934-1:2016 Non-destructive testing. Magnetic particle testing. Part 1: General principles) and periodically stopping the test and disassembling the threaded joint.

After the first measurements, the threaded joint was reassembled and re-tightened. Each time the position of the nuts had to be the same as during the cyclic strain process prior to disassembly of the joint. This was achieved using auxiliary nuts and pins (7) that hold the stud (1) in position relative to the nuts (2). The calculated force of the first loading cycles was determined using the tension variation cycle. Further strain was carried out according to the displacement registered during the first cycles. The testing machine was then stopped to measure the crack and register the force of the last cycle. Following measurement, the nut–stud–nut system was reassembled, and further application of the strain was resumed starting with the force registered. Then, it was moved to the fixed displacements. The maximum and minimum cycle load was measured during crack propagation.

### 3.2. Influence of Tightening and Cyclic Bending

An investigation of the patterns of force distribution was carried out according to the testing scheme depicted in Figure 2. The loading modes of the investigated threaded joints were (1) $\sigma = (0.6 \pm 0.35)\sigma_y$ – 10 cycles; and (2) $\sigma = (0.6 \pm 0.45)\sigma_y$ – 200 cycles. The force distribution after tightening the threaded joint to $\sigma_t = 0.6\sigma_y$, as seen in Figure 3, corresponded to the characteristic general pattern of the joints subjected to tension (tensioned stud and compressed nut). The strains that occurred in the tensioned layers due to

bending in the turn cavities during the first load cycle also corresponded to the general patterns of force distribution of the tensioned joints. The general stress and nut effect in the compressed layers caused distortion in the known patterns. The strains of the tensioned region of the stud in the turn cavity located at the pitch distance from the nut are also shown in Figure 3, which depicts a free turn. The experiment showed that the increase in stress caused by bending from $0.35\sigma_y$ to $0.45\sigma_y$ proportionately did not lead to an increase in forces in turns 1 and 2 of the threaded joint.

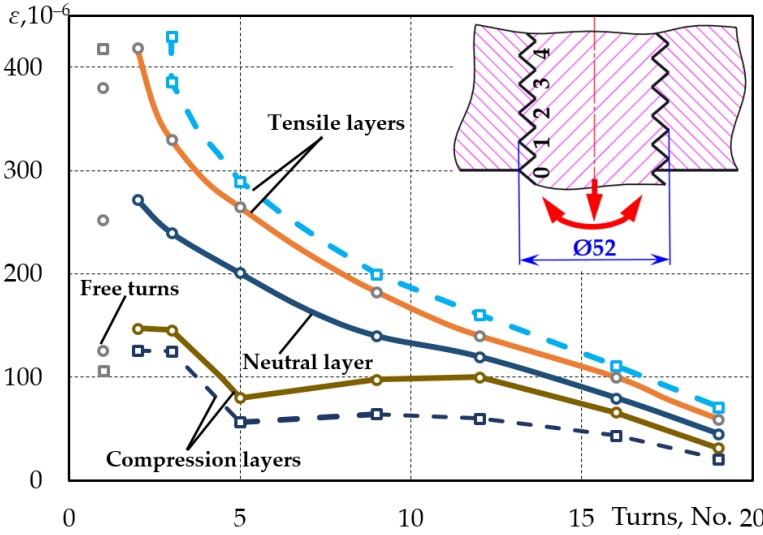

**Figure 3.** Strains in individual turns of the stud M52×4: $\bigcirc$—$\sigma_t = (0.6 \pm 0.35)\sigma_y$; $\square$—$\sigma_t = (0.6 \pm 0.45)\sigma_y$.

Force distribution patterns did not change during further application of strain up to 200 cycles. Maximum stress was obtained in the bending plane regardless of the position of the nut–stud with respect to each other. However, the position of the nuts had an effect on the conditions of initiation and propagation of the crack. In view of the force distribution, one of the nuts was placed in the appropriate position, while another was positioned according to the tightening value. The value of looseness was important for both individual and group joints, as it led to lower tightening of the connected parts and a change in the loading conditions on the assembly. Figure 4 presents the experimental data for the threaded joint. Tension stress in the tested threaded joints varied from $\sigma_t = 0.95\sigma_y$ to $\sigma_t = 1.2\sigma_y$. The maximum tension stress values caused by the bending varied from $\sigma_b = 0.35\sigma_y$ to $\sigma_b = 0.55\sigma_y$. In the case of initial stress $\sigma_t > 0.95\sigma_y$, a more pronounced reduction in stress was observed at the beginning of the cyclical strain. In the case of a strong tightening $\sigma_t > \sigma_y$ (curve 3, Figure 4) and single-time stud tension during joint assembly, stress was considerably reduced during the first cycles. Further cyclic bending to crack initiation was virtually insignificant for the stability of tightening. Residual strain was recorded in the failure layers, outlying layers. The durability of the studs was reduced both in terms of crack initiation and in the terms of final failure in case of the same stress value and increasing bending stress.

With the pitch increasing, the stress was reducing faster during the first load cycles. With the crack propagating, a more considerable reduction in the stress would not be observed in the case of increase in the pitch. The stress of the stud began to gradually reduce in the case of crack propagation (Figure 4, curves 1' and 2'). When the initial stress was $\sigma_t \approx \sigma_y$, the tightening was considerably reduced during the first cycles. Subsequently, the stress remained stable during crack initiation, and the curves straightened out. The strains increased in their environment only after the crack had reached a certain value, while the stress reduced. The experiment showed that the increase in stress caused by bending from $0.35\sigma_y$ to $0.45\sigma_y$ proportionately did not lead to an increase in forces in turns 1 and 2 of the threaded joint. The force distribution patterns did not change during further application of strain up to 200 cycles. The testing machine was then stopped to

measure the crack and register the force of the last cycle. Following the measurement, the nu—stud–nut system was re-assembled, and further application of strain was resumed starting with the force registered. It was then moved to the fixed displacements. The loading force was determined periodically during cyclic strain. The crack was measured by the magnetic luminescent method. The maximum and minimum cycle loads were measured during the crack propagation (this was difficult to do during the last cycles, with the crack approaching the critical value). The test results, i.e., the variation of the maximum stress depending on the number of cycles and coefficient of asymmetry, are presented in Figure 5.

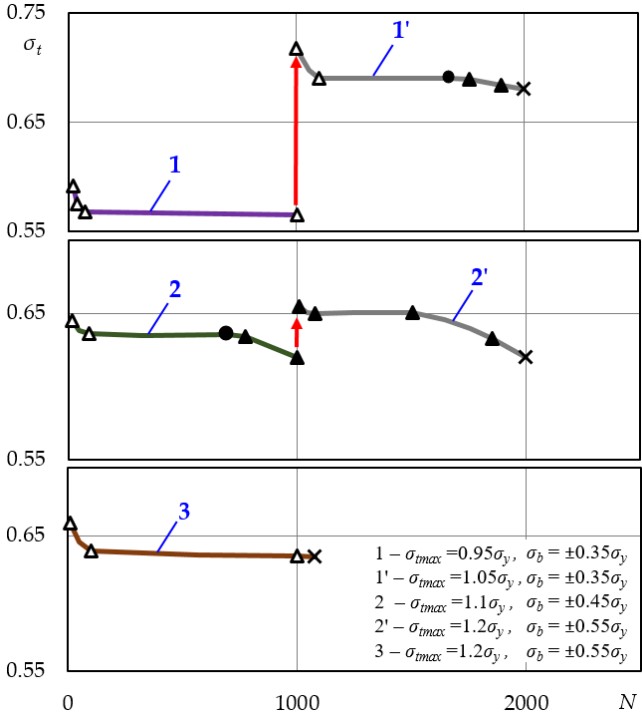

**Figure 4.** Variation of the stress of the threaded joints M52×4: Δ—measurement limits; ▲—measurement limits upon crack initiation; ●—crack initiation; ✕—testing stopped.

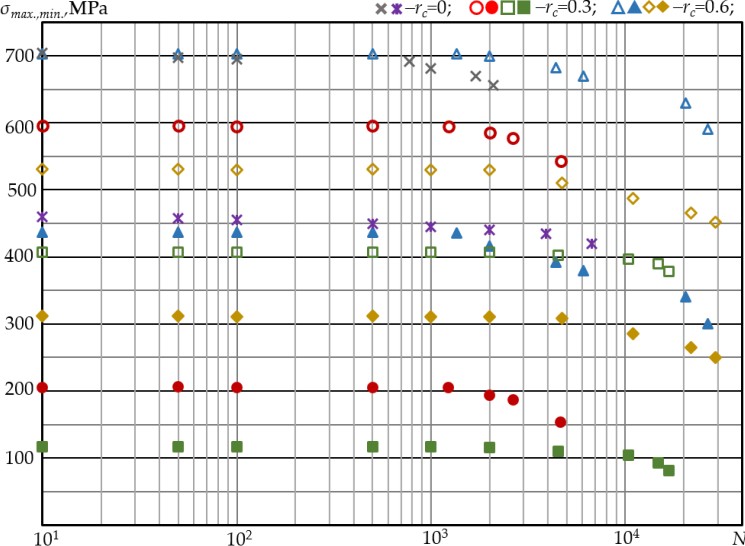

**Figure 5.** Variation of the stresses in the M52×4 stud depending on the number of cycles and coefficient of asymmetry; the empty symbols refer to maximum cycle stresses and the filled symbols represent the minimum stresses.

### 3.3. Comparison of the Experimental Results to Benchmark Curves

In the normative calculations for the nuclear energy installations, the low-cycle durability calculations were used to establish the actual margin and compare it with the regulated or desired value. According to PNAE norms, the mean stress in the cross-section caused by the internal pressure, tightening, and thermal forces shall not exceed $0.65\sigma_y$, or $0.85\sigma_y$ in view of bending. Stress caused by internal pressure may exceed the analog stresses permissible in ASME norms. The mean stresses caused by all loads are virtually the same, while PNAE establishes stricter limitations on axial and bending stresses compared to ASME norms. The benchmark fatigue curves were obtained according to the PNAE norms where the strain concentration factor is equal to 4 (margin $n_\sigma = 1.5$; $n_N = 3$) for the metric thread in the tensile area, and, for the elastic-plastic area, it could be calculated according to Neuber's rule [34] (margin $n_\sigma = 1.5$; $n_N = 5$), reaching 5.5.

According to the PNAE norms, the admissible number of cycles $N_{adm}$ is determined at a certain amplitude of the stress cycle, and an admissible stress amplitude $\sigma_{a,adm}$ is determined by providing a number of cycles. This is implemented using two methods: according to the calculated fatigue curves provided in the norms and according to the formulas, if the number of cycles does not exceed $10^6$. In the PNAE normative methodology, low-cycle durability is determined according to the first stage of failure, i.e., crack initiation in the stress concentration areas.

The crack propagation stage was not considered due to the increased safety requirements. The calculation was based on binomial-type formulas [35–38]; the terms whereof expressed the plastic and elastic components of the strain amplitude:

$$
\begin{aligned}
\sigma^*_{a,adm} &= \frac{E \cdot e_c}{n_\sigma (4 \cdot N_{adm})^m} + \frac{\sigma_{-1}}{n_\sigma \left(1 + \frac{\sigma_{-1}}{\sigma_u} \cdot \frac{1+r_a}{1-r_a}\right)} \\
\sigma^*_{a,adm} &= \frac{E \cdot e_c}{n_\sigma (4 \cdot n_N \cdot N_{adm})^m} + \frac{\sigma_{-1}}{1 + \frac{\sigma_{-1}}{\sigma_u} \cdot \frac{1+r_a}{1-r_a}}
\end{aligned}
\tag{1}
$$

The smaller value of the number of load cycles to crack initiation in the stud $N_0$ was determined on the basis of the two values calculated using these formulas and was considered to be a calculation result. According to the ASME code, in the case of the high-strength studs (700÷1150 MPa), the low-cycle durability is determined according to the calculated durability curves that are verified experimentally. For the threaded joints, the curves were calculated using the formula [39–41]:

$$
\sigma^*_{a,adm} = \frac{A_e \times E}{N_f \times C} + S_e
\tag{2}
$$

Figure 6 presents a comparison of the experimental results for the joints bolted using crack initiation with the fatigue curves calculated according to the norms.

It could be seen that the experimental values of cyclic durability were considerably higher (2- to 10-fold) than the values reflected by the norm curves.

Individual stud and bolt failure stages are not identified in ASME standards, and the safety factors are fairly large. In the case of calculation using the nominal stresses caused by internal pressure, the fatigue safety factors according to the PNAE norms are not as conservative as in the case of ASME norms. However, a direct comparison of fatigue curves, where the number of cycles does not exceed $10^8$, including an increase in concentration factors, shows that the permissible amplitude and number of cycles are subject to stricter regulation in the PNAE norms than in the ASME norms. The total tensile and bending stresses in the outlying layers of the bolt may sometimes exceed the yield strength. The resulting nonelastic strain conditions along the entire length of the bolt may lead to changes in the operating conditions of the joint. Nonplastic strain processes determine the failure, which may be a localized or general failure. In the first case, this is related to the accumulation of damage under the action of cyclic loads that lead to the propagation of the crack.

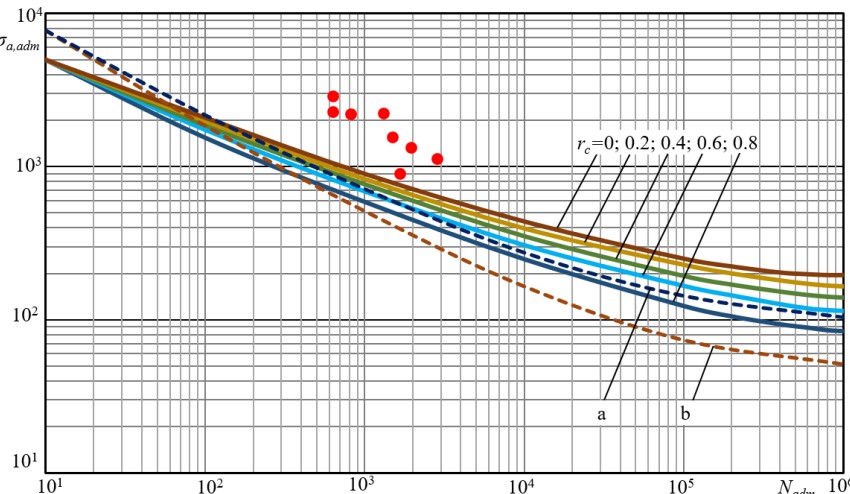

**Figure 6.** Comparison of the experimental results with the fatigue curves according to the PNAE (straight line) and ASME (dotted line) norms: •—experimental data; a—maximum nominal stresses $0.95\sigma_y$; b—maximum nominal stresses $0.6\sigma_y$.

In the second case, the failure is determined by the accumulation of relative displacements. A progressive geometry change of the bolt is undesirable during operation. It may cause a change in the tightening, redistribution of forces, and leak tightness of the connected elements.

### 3.4. Analysis of the Crack Propagation

The crack initiation and propagation patterns in the tightened threaded joint subjected to cyclic bending had specific attributes compared to the joints subjected to cyclic tension. Distribution of the stresses was non-uniform along the turn. The turn load increased closer to the tensioned layers and reduced when positioned further away from them. In the stud–nut force distribution system, the investigation showed the position subjected to the highest load. The maximum stresses resulting from the total stress and bending action developed only on the strain plane. This determines the stress–strain state and crack initiation conditions in the root of the turn. The crack initiation and propagation analysis showed that the crack may be either one- or two-sided (Figure 7).

When the crack propagates in the stud cross-section, as in Figure 7a, it is either one-sided or asymmetric. When the crack reaches the critical value, the remaining part fails largely due to the strain energy accumulated during stud tension.

If the cross-section of the crack occurs in the cross-section of the stud, as shown in Figure 7b, it is symmetrically two-sided. When the crack reaches the critical value $h_{cr}$ or $h_a = 2h_{cr}$, the rest of the crack disintegrates, mainly due to the strain energy that has been accumulated by stretching the stud. In contrast, cyclic crack propagation is due to bending deformation.

The study of the distribution of the load in the stud–nut system showed the position of the most loaded position, so that when the threaded joint was assembled, one of the nuts was placed in such a way that the stud received a uniform load. The other nut can be placed in any position depending on the amount of tension (Figure 2a). The combined stress and bending action of the stud results in maximum indentations only in the plane of deformation. This determines the stress–strain state and the conditions for the formation of a crack in the cavity of the turns.

Investigating the stress behavior and experimental studies of the indentation distribution show that, depending on the position of the nut in the bending plane, the angle $\alpha$ can vary from $0°$ to $60°$ depending on the position of the nut in the maximum stresses in the thread recess. Patterns of crack initiation and propagation also vary. Fracture analysis showed that the angle of initiation of the crack in the recess varies from $0°$ to $45°$. The

initiation of the fracture determines its subsequent propagation according to one of the patterns shown in Figure 7c.

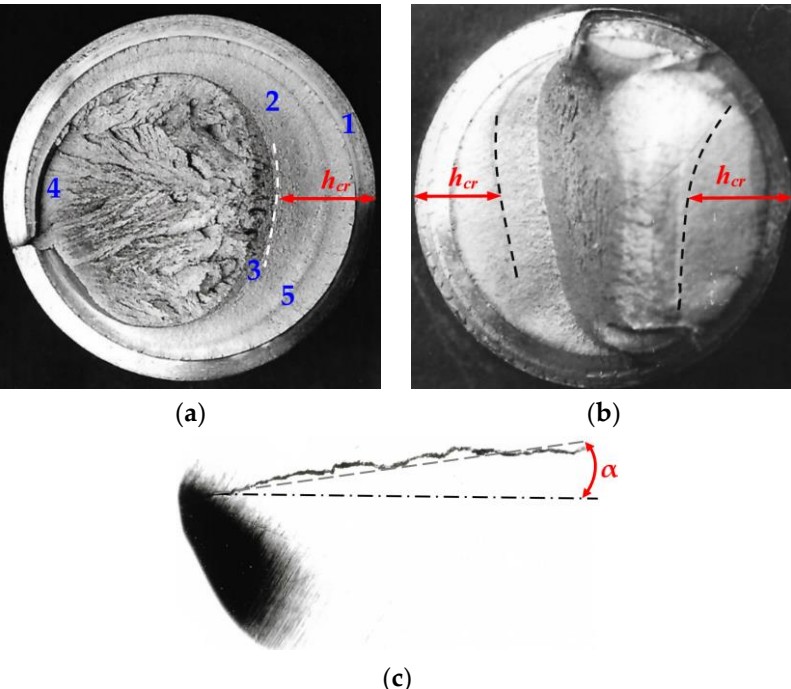

(a)　　　　　　　　　　　　　　　　　　　(b)

(c)

**Figure 7.** (**a**)—one-sided fracture of the stud: 1—crack area; 2—normal propagation area of the crack; 3—accelerated propagation area of the crack; 4—brittle failure area; 5—trace of sample stoppage; (**b**)—two-sided fracture of the stud; (**c**)—the start of crack propagation in the root of the stud thread profile.

### 3.5. Loosening Analysis

The loosening of tension and bending connections is almost unexplored in scientific papers. The tightening forces of the structural members connected by threaded connections are an important factor in the reliability of an assembled unit or device. However, under cyclic loads, threaded connections loosen, which can lead to a loss of tightness of a heavily stressed vessel or to the failure of a piece of equipment. However, under cyclic loads, the threaded connections loosen, which can lead to a tightness failure of a heavily pressured vessel or to the failure of a component in the operational condition of the device. Due to the clamping of the contact surfaces between the flanges and the thread, the self-rotation of the nut, the plastic deformation in the thread and the flanges, the tension of the bolt and the tightness of the flanges may be reduced. The tensioning force must be measured for any method of tightening a threaded joint. In the experiments described in Section 3.1, it was observed that the stress of the studs decreased significantly after the threaded connections were assembled. Figure 8 shows the stress of the variation of the stud stress during the assembly of the threaded joint in the bending fixture.

During the first clamping cycle, the tightening stresses were already reduced by $\Delta\sigma$ when the stud was tensioned and the intermediate parts were compressed simultaneously (Figure 8a). The stud was repeatedly tensioned and the spacers were compressed to achieve the desired tightness. The amount of deformation of the stud was then measured by turning the nut and removing the external tension. As shown in Figure 8b, the stresses in the stud were reduced by an amount $\Delta\sigma$.

It was only after 3–4 repetitions of the tightening process that the desired stud tension and interference fit were achieved. If the tightening was limited to a single tightening, a significant (up to 16%) loosening of the threaded joint would be observed at the beginning of the cyclic bending. The loosening of threaded joints during cyclic deformation is characterised by the load (stress $\sigma$—strain $\varepsilon$) diagrams of threaded joints based on experimental

data. Figure 9a–c provide a diagram of the stress variation in the bending plane of the stud edge layers.

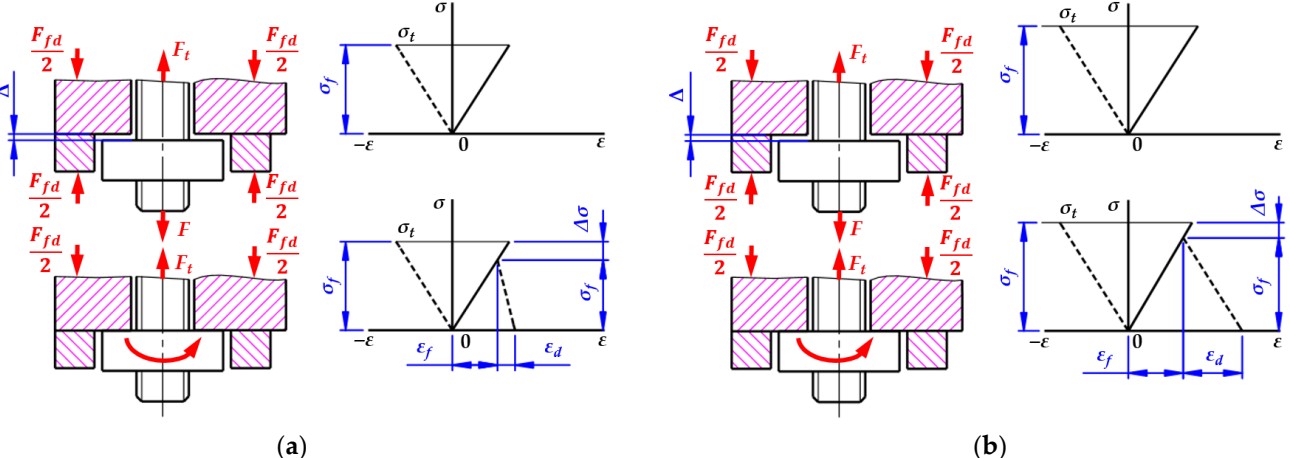

**(a)**　　　　　　　　　　　　　　　　　　　　　　　　**(b)**

**Figure 8.** Tension stress changes (decreasing) during assembly of the threaded joint: (**a**)—initial tightening cycle; (**b**)—repeated tightening cycles.

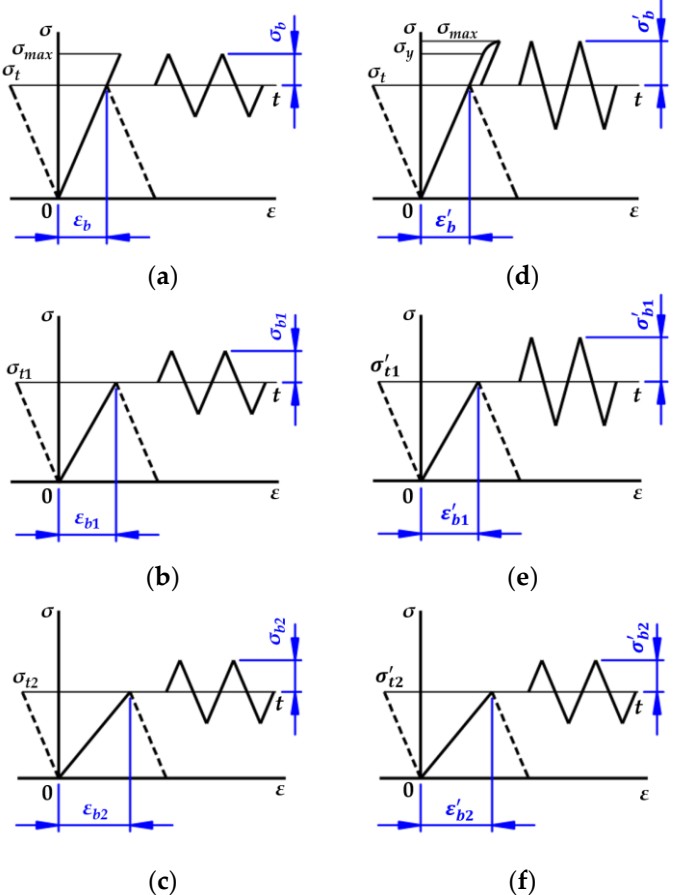

**(a)**　　　　　　**(d)**

**(b)**　　　　　　**(e)**

**(c)**　　　　　　**(f)**

**Figure 9.** Loading diagrams of the threaded joints in the bending plane: when $\sigma_t < \sigma_y$ (**a**–**c**); when $\sigma_t > \sigma_y$ (**d**–**f**).

If, in the elastic phase, the maximum stresses did not exceed $\sigma_t < \sigma_y$, then after 50 to 100 cycles the initial stress decreased to $\sigma_{t1}$ and the bending stresses were almost unchanged $\sigma_{b1} \approx \sigma_b$. As the crack spread, the tension decreased to $\sigma_{t2}$ just before the fracture and the bending stresses to $\sigma_{b2}$. If the stresses in the relative elasticity stage exceeded the yield stress $\sigma_t > \sigma_y$ (Figure 9d–f), then after 50 to 100 cycles the initial stress decreased to $\sigma'_{t1}$,

and the bending stresses were almost unchanged at $\sigma'_{b1} \approx \sigma'_b$. As the crack spread to the fracture, the stress decreased to $\sigma'_{t2}$, and the bending stresses decreased to $\sigma'_{b1}$.

Figure 10 shows the tension reduction in the stud in a plane perpendicular to the bending (this value was continuously monitored by two sensors). After joint assembly, the stud stress was $\sigma_t$. After 50 to 100 cycles of stress-controlled cyclic bending, the stud t decreased by the amount $\Delta\sigma_t = \sigma_t - \sigma_{t1}$. For a certain number of cycles before the crack occurs, the tension was almost constant. As the crack spread, the strain decreased, and the difference just before the fracture compared to a stable process was $\sigma'_t = \sigma_{t1} - \sigma_{t2}$.

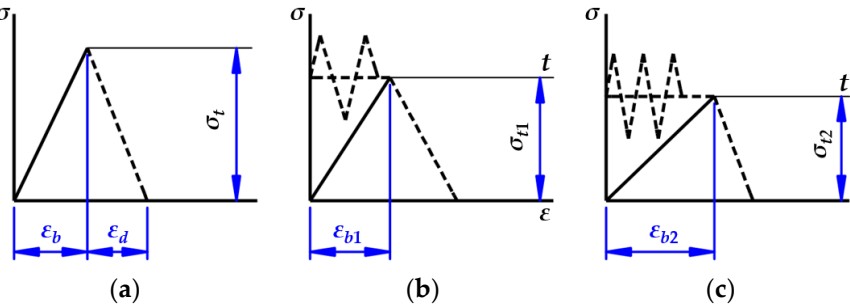

**Figure 10.** Loading diagrams of threaded joints in the plane perpendicular to bending: (**a**)—initial tightening cycle; (**b**)—at the start of cyclic loading; (**c**)—before fracture.

No self-turning of the nut was observed in the tested connections. All threaded joints of the appropriate diameter were tested with the same device, so the deformation of the contact surfaces of the intermediate parts is almost irrelevant to the results obtained. The variation in stud stress during the loading with a controlled strain ($\varepsilon$ = const.) cyclic bending process is related to the crimping of the contact surfaces of the stud and the nut, the shear of the first turns, the bearing surface of the crimping of the bearing surface of the nut, and the deformation in the gap environment. The latter factor must be distinguished from the others as it has the greatest influence. From the experimental data obtained, it can be concluded that, if the assembly has been carried out by multiple turns of the nut, the stress on the stud at the beginning of the load is reduced by a maximum of 5% for $\sigma_t > \sigma_y$ and by about 3% for $\sigma_t < \sigma_y$. In the general case, the relaxation of the nut–stud–nut system is made up of several components:

$$\Delta\sigma_t = \Delta\sigma_{tb} + \Delta\sigma_{tn} + \Delta\sigma_{td} + \Delta\sigma_t^{\circlearrowright} \tag{3}$$

Based on the experimental data obtained by the authors, a relationship between the stud tension stress and the size of the crack, expressed as $h_{cr}/d$ ($h_{cr}$ is the depth of the crack, $d$ is the diameter of the stud shank), was established, as shown in Figure 11.

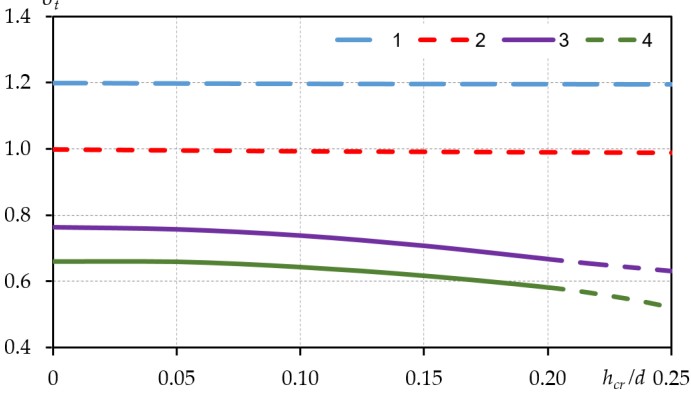

**Figure 11.** Tension stress dependence of crack parameters.

In the relatively elastic stage, the stresses due to tension and bending at the beginning of the deformation are represented by dashed lines 1 and 2. After the first few cycles, the stress decreased from $0.8\sigma_y$ to $0.76\sigma_y$ and remained stable for a long time. The stress magnitude as the crack started and spread is shown in curve 3. If the initial stress was $0.7\sigma_y$, the magnitude of the stress varied with the crack according to curve 4.

This chapter presents a study on the tension stress in threaded joints during the different phases of cyclic loading, starting from the assembly. By tightening the stud and multiple turns of the nut, it is possible to avoid torsional stresses (which significantly reduce the strength of the stud) and stabilize the tension already in the tightening process. The following failure process parameters were obtained and used in the next section to analyse the shakedown of the threaded joints: the geometry of the crack, its propagation patterns, and the magnitude of the stress.

## 4. Shakedown of the Threaded Joints

Due to the considerable mechanical and thermal effect, the element of the structure may show low-cycle fatigue cracks, and residual strain may accumulate. In this case, the parameters of the variable nonelastic strain process limit the designed durability. If the residual strain resulting from the plastic flow at the start of loading no longer accumulates after a certain number of cycles, this means that the structure has adapted to the load. The sufficient shakedown conditions are based on the E. Melan [42,43] principle (static shakedown theorem) and W. T. Koiter [44] kinematic theorem. The experimental investigation conducted on threaded joints showed that in certain cases, the total stress caused by the tightening and cyclic bending reaches the plastic area and penetrates to a certain depth. Further repetitive variable loading affects the strength and reliability of the joint. It was noticed that the cyclic plastic strain no longer accumulates, i.e., the joint shakedown takes place as soon as a favorable field of residual stress has developed after a small number of cycles. In case of violation of the shakedown conditions, plastic strain with a variable sign (usually local) may occur, or single-sign strain may grow in each cycle of the structure or a part thereof. This results in a progressive shape change that covers the entire structural element or its part. Theoretical papers on the topic of shakedown provide a revision of the cyclic strength margin of threaded joints.

The alternating plastic strain appears where, at any moment in the cycle, there are no constant stresses, the total amount of which together with the stresses relatively proportionate (according to Hooke's law) to the external action would be less than the threshold stresses $\sigma_s$, at least at one point in the structure. If the stresses in the structure are proportionate to only one parameter, plastic strains with variable sign start developing, where the proportional strain interval exceeds $2\sigma_s$.

In case of a change of conditions, the alternating plastic strain may not develop. Instead, the progressive shape change may appear in cases where constant stresses (necessary for the total stresses at each point of the body at any moment of the cycle, without variation of the loads) do not exceed the threshold stresses as they do not satisfy the equilibrium conditions (with zero loads in individual cases, where no constant external forces are present).

This is the static assessment of the bottom threshold of the progressive shape change. The upper threshold may be analyzed from the kinematic approach; the progressive shape change will take place in the case of the distribution of the increment of irreversible strain, at which the action of the minimum threshold stresses of the cycle and tensile stresses caused by the difference of the external action is not positive and calculated for the entire volume of the structural element.

The threshold values of the external parameters can be determined by analyzing the shaking conditions. In case the values are slightly exceeded, this leads to the increase in the interval of nonelastic strains and their incremental growth per cycle. The normal action of the structure occurs where the plastic strain does not vary during stationary loading. The shaking conditions of elements whose durability is determined by a fairly small number

of load variation cycles may be used as a failure criterion. Experimental and theoretical investigations of threaded joints subjected to tension and bending may be used to complete the assessment of the possibility of shakedown.

### 4.1. Crack-Free Joints

The progressive shape change conditions in the case of repetitive variable loading are determined according to the elastic–plastic body shakedown theories and the experimental data. W. T. Koiter [44] formulated the kinematic theorem (two statements) in terms of kinematic definitions that determine the work of external forces and the dissipation of plastic energy over the cycle time [45] (p. 25):

$$
\begin{aligned}
\int_0^T \left\{ \int_V x_i^0 u_{i0} dV + \int_{A_p} p_i^0 u_{i0} dA \right\} dt &\geq \int_0^T dt \int_V F(\varepsilon_{ij0}'') dV \\
\int_0^T \left\{ \int_V x_i^0 u_{i0} dV + \int_{A_p} p_i^0 u_{i0} dA \right\} dt &< \int_0^T dt \int_V F(\varepsilon_{ij0}'') dV
\end{aligned}
\tag{4}
$$

Gokhfeld and Cherniavsky [45] (p. 33) proposed representing plastic energy dissipative function $F(\varepsilon_{ij0}'')$ as follows:

$$
\int_0^T dt \int_V \min_t \left[ (\sigma_{ij} - \sigma_{ij}^{(e)}) \varepsilon_{ij0}'' \right] dV
\tag{5}
$$

Following the previous one, it can be written as:

- a given structure shakedown:

$$
\int_0^T dt \int_V (\sigma_{ij} - \sigma_{ij}^{(e)}) \varepsilon_{ij0}'' dV > 0
\tag{6}
$$

- a given structure that does not undergo shakedown, i.e., the structure must collapse eventually due to cyclic plastic deformations:

$$
\int_0^T dt \int_V (\sigma_{ij} - \sigma_{ij}^{(e)}) \varepsilon_{ij0}'' dV < 0
\tag{7}
$$

Therefore, the elastic-plastic body shakedown condition will be as follows:

$$
\int_0^T \left\{ \int_V x_i^0 u_{i0} dV + \int_{A_p} p_i^0 u_{i0} dA \right\} dt < \int_0^T dt \int (\sigma_{ij} - \sigma_{ijt}^{(e)}) \varepsilon_{ij0}''
\tag{8}
$$

The kinematic shakedown theorem is based on the fundamental assumption of the admissible cycle of plastic strain rates. According to the definition of the plastic strain increment in such a cycle over a specified time interval:

$$
\Delta \varepsilon_{ij0}'' = \int_0^T \varepsilon_{ij0}'' dt
\tag{9}
$$

The increments of residual translations:

$$
\Delta u_{i0} = \int_0^T u_{i0} dt
\tag{10}
$$

Considering that the integration over time and over the volume of the body (or over its surface) is mutually independent:

$$\int_0^T dt \int_V x_i^0 u_{i0} dV = \int_V x_i^0 \Delta u_{i0} dV \tag{11}$$

$$\int_0^T dt \int_{A_p} p_i^0 u_{i0} dA = \int_{A_p} p_i^0 \Delta u_{i0} dA \tag{12}$$

Minimum values of continuous stresses:

$$\sigma_{ij}^0 = \sigma_{ij} - \sigma_{ij}^{(e)} \tag{13}$$

Equation (8) can be written as follows:

$$\int_V x_i^0 \Delta u_{i0} dV + \int_{A_p} p_i^0 \Delta u_{i0} dA < \int_V \sigma_{ij*}^0 \Delta \varepsilon_{ij0}'' dV \tag{14}$$

The shakedown theory widely uses approximate methods. The approximate kinematic method enables an approximation of the true failure mechanism and a reduction in the error of the threshold cycle parameters. The increments in plastic strain are determined according to the potential nature of the failure mechanism and the selected redistribution of displacements $\Delta u_{i0}$. The respective fictive yield stresses $\sigma_{ij*}^0$ are determined by using the associated flow rule. According to the experimental data, crack initiation and propagation take place in the weakest link of the threaded joint, i.e., in the stud, after a certain number of cycles. Plastic strains occur in the outer layers of the stud before crack initiation, under the action of repetitive variable load. As a rod with circular cross-section, the stud is subjected to axial force $F$ and symmetrically variable bending moment $M$ ($-M^* < M_f < M^*$). Figure 12 shows a cross-section of a stud and the stress distribution under $M$ and $F$ forces leading to yield stresses ($\sigma_y$—yield stress). The tension zone $A$ is equal to the compression zone $C$. Consequently, the longitudinal force $F$ is equilibrated to the stresses acting in zone $B$. Bending moment $M$ equilibrates with the stresses acting in zones $B$ and $C$.

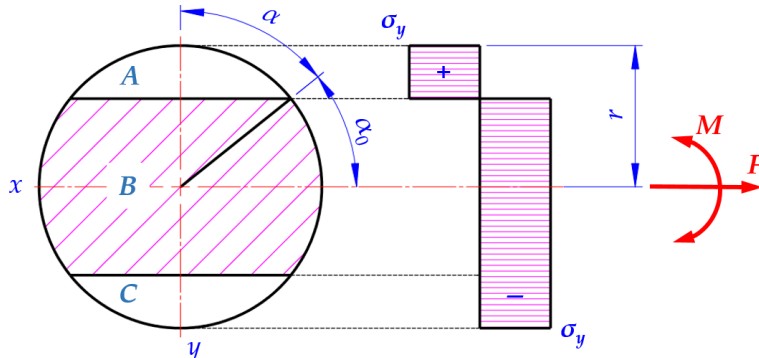

**Figure 12.** Cross-section of the stud and stress distribution under $M$ and $F$ forces.

In the task analyzed $\sigma_{ij*}^0 = \sigma_z$ it is assumed that the failure mechanism is a uniform elongation of the stud:

$$\Delta u_{i0}(x, y, z) = \Delta u_{z0} = constant \tag{15}$$

Respectively:

$$\Delta \varepsilon_{ij0}'' = \Delta \varepsilon_{z0}'' = \frac{\Delta u_{z0}}{l}, \text{ and } p_i^0 = p_z^0 = \frac{4F}{\pi d^2} \tag{16}$$

Whereas:

$$\sigma_z^{(e)} = \frac{M}{I_x}y \tag{17}$$

Since bending moment $M$ varies according to a symmetric cycle and $\Delta\varepsilon_{z0}'' > 0$, one can be obtained:

$$\sigma_{ij*}^0 = \begin{cases} \sigma_y - \frac{M}{I_x}y, \frac{d}{2} \geq y \geq 0 \\ \sigma_y + \frac{M}{I_x}y, 0 \geq y \geq -\frac{d}{2} \end{cases} \tag{18}$$

Plastic energy dissipation:

$$D = \int_V \sigma_{ij*}^0 \Delta\varepsilon_{ij0}'' dV = 4 \int_0^{l_0} dz \int_0^{d/2} dx \int_0^{\sqrt{\frac{d^2}{4}-x^2}} (\sigma_y - \frac{M}{I_x}y)\frac{\Delta u_{z0}}{l_0}dy \tag{19}$$

The progressive shape change condition can be written down as follows:

$$F \cdot \Delta u_{z0} = 4\Delta u_{z0}\left[\sigma_y\frac{\pi d^2}{16} - \frac{1}{2}\frac{M}{I_x}\frac{d^3}{12}\right] \tag{20}$$

Dimensionless loading parameters $n = F/F_y$ and $m = M/M_y$ are introduced with the threshold parameter values:

$$F_y = \sigma_y\frac{\pi d^2}{4} = \sigma_y \cdot \pi \cdot r^2, M_y = \sigma_y\frac{d^3}{6} = \sigma_y\frac{(2r)^3}{6} \tag{21}$$

After that, these are entered into Equation (20), which results in the following expression of the progressive shape change:

$$n + \frac{64}{9\pi^2}m = 1 \text{ or } an + bm = 1 \tag{22}$$

Plastic failure is obtained by comparing the values of components of variable stresses with the yield stress:

$$2\frac{M}{I_x}\frac{d}{2} = 2\sigma_y \tag{23}$$

After the dimensionless parameters are entered, the following is obtained:

$$\frac{32}{6\pi}m = 1 \tag{24}$$

Plastic failure is unreal in threaded joints; hence, only the progressive shape change condition is analysed in the present work. The shakedown diagram of the threaded joints M52×4 was designed according to dependencies (22) and (24). Figure 13 shows the shakedown diagram built according to [46] (p. 42) and the experimental data according to the established pattern, that is, after about 50 cycles, when the stress is constant in the joints.

The coordinates analyzed (point $D$) are $n_d$ and $m_d$, if the threaded joint is subject to axial force $F$ and bending moment $M$. Limit point $L$ coordinates $m_{\lim.}$, $n_{\lim}$. The inclination angle of the radius for similar cycles is

$$tg\beta = \frac{m_{\lim}}{n_{\lim}} \text{ or } tg\beta = \frac{m_d}{n_d} \tag{25}$$

The safety factor of the progressive shape change is the ratio:

$$\eta = \frac{m_{\lim}}{n_{\lim}} \text{ or } \eta = \frac{m_d}{n_d} \tag{26}$$

Using Equation (22), it could be written down that:

$$\eta = \frac{1}{an_d + bm_d} \tag{27}$$

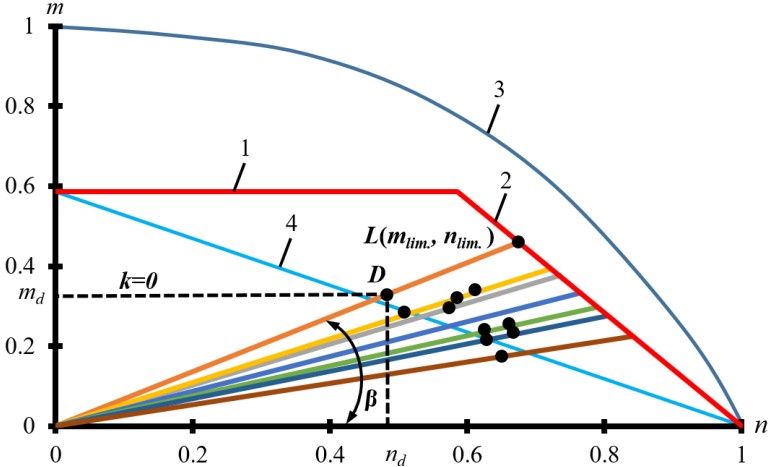

**Figure 13.** Position of the threaded joints in the shakedown diagram: 1—variable flow condition; 2—progressive shape change condition; 3—collapse range; 4—elastic range; •—experimental data.

The methods used for low-cycle fatigue could be used for the statistical evaluation [47,48] of progressive shape change. The workpieces used in the production of the threaded joints are made of the same alloy and use the same heat treatment. Indicators of the static mechanical properties ($\sigma_y$ = 930 ÷ 1010 MPa) were determined for each of them. These were used to select the tightening and cyclic loading conditions. The differences in mechanical properties suggest that the metal contained defects and that the metallurgical and heat treatment processes resulted in a heterogeneous structure. This led to non-uniform distribution of the hazard level of the structure across the volume of the structure. This indicates the statistical nature of the low-cycle fatigue of threaded joints, i.e., for the reasons of the distribution of fatigue-related factors. The respective assumptions also emerge due to differences in certain test conditions, such as deviation from the ideal specimen, machine axiality, variations in the load level, action of the nut on the bolt, and other factors. The experimental data for the threaded joints were statistically analyzed according to the normal distribution law:

$$P(x) = \frac{1}{\sigma_0 \sqrt{2\pi}} \exp\left[ -\frac{(x_i - x_0)^2}{2\sigma_0{}^2} \right] \tag{28}$$

The mean square deviation was calculated according to the following:

$$\sigma_0 = \sqrt{\frac{1}{n_0 - 1} \sum_{i=1}^{n_0} (x_i - x_0)^2} \tag{29}$$

Mean arithmetic value:

$$x_0 = \frac{1}{n_0} \sum_{i=1}^{n_0} x_i \tag{30}$$

Dependence used for the calculation of the random value probability:

$$P = \frac{m_0 - 0.5}{n_0} \tag{31}$$

The safety factor was calculated using Equation (27), where the probability ranged from 1% to 99%, changing from 1062 to 1689. At 50% probability, the safety factor was

$\eta = 1322$. The developed probabilistic dependencies on the coordinates $P$-$n_d$ and $P$-$m_d$ are presented in Figure 14.

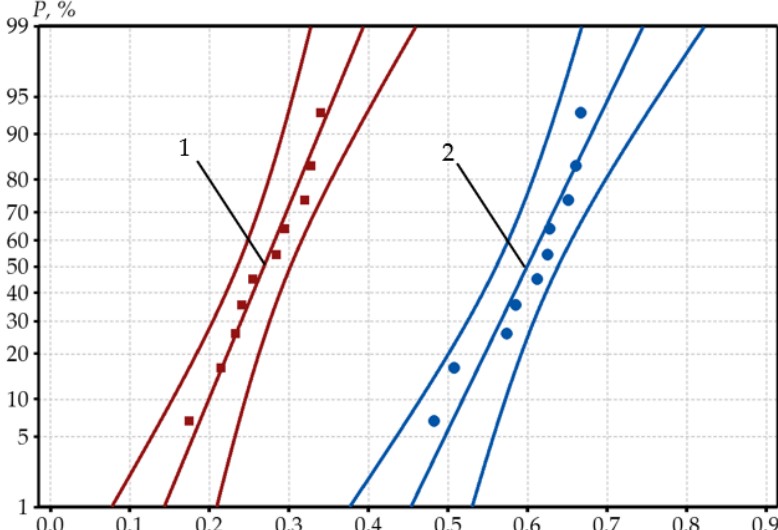

**Figure 14.** Probabilistic relationships of the parameters $m_d$ (1) and $n_d$ (2) and their 95% probabilistic intervals for crack-free threaded joints.

The hypothetical probabilistic curve obtained was used to determine the probabilistic values for the parameters $n$ and $m$, where $P = 1\%, 10\%, 30\%, 50\%, 70\%, 90\%$, and $99\%$. The results obtained are depicted in the shakedown diagram (Figure 15).

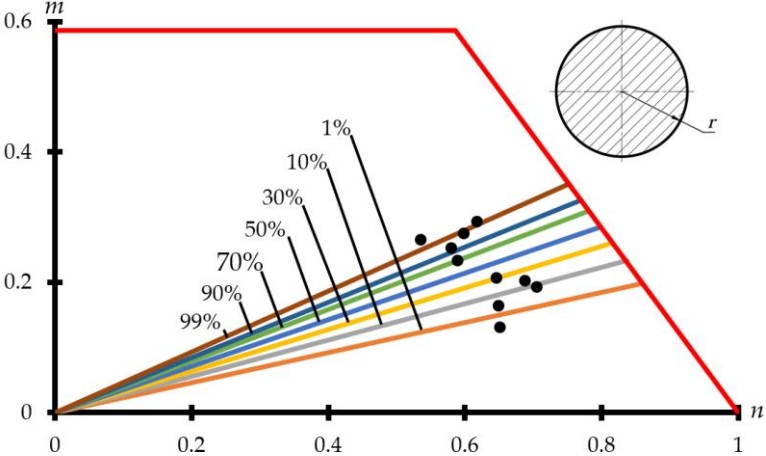

**Figure 15.** Statistical assessment of crack-free threaded joints: ●—experimental data.

### 4.2. Joints with a One-Sided Crack

The experimental investigation showed that crack initiation and propagation only had specific characteristics of only threaded joints in the case that the studs were deformed, as depicted in Figure 16. When the stud was subjected to the stress and bending moment variable according to the symmetrical cycle, the crack was one-sided in the case of a certain position of the nut.

In this case, according to the assumption of ideal elasticity, the variable components of the stresses were calculated using the following equation:

$$\sigma_z^{(e)} = \frac{M}{I_{x1}}(y + y_c), \quad -r \cdot \cos\alpha \leq y \leq r \cdot \cos\alpha \tag{32}$$

For the fictive yield surface, the stresses were calculated using the following equations:

$$\sigma_{ij}^0 = \begin{cases} \sigma_y - \dfrac{M}{I_{x1}}(y + y_c), 0 \leq y \leq r\cos\alpha \\ \sigma_y + \dfrac{M}{I_{x1}}(y - y_c), -r\cos\alpha \leq y \leq 0 \end{cases} \tag{33}$$

The geometric indicators of the cross-section with a one-sided crack were calculated according to Figure 16. The progressive shape-change condition can be written down as follows:

$$F \cdot \Delta u_{z0} = 4\int\limits_0^{l_0} dz \int\limits_0^{r\cdot\cos\alpha} dy \cdot \int\limits_0^{\sqrt{r^2-y^2}} \left[\sigma_y - \frac{M}{I_{x1}}(y + y_c)\right]\frac{\Delta u_{z0}}{l_0}dx \tag{34}$$

By the integration of Equation (34) and the introduction of dimensionless load parameters $m$, $n$, the following was obtained:

$$\frac{\pi}{\pi + \sin 2\alpha - 2\alpha}n + \frac{16r^4}{3I_{x1}}\left[\frac{\sin^3\alpha}{6(\pi - \alpha + \sin\alpha\cos\alpha)} + \frac{1 - \sin^3\alpha}{3(\pi + \sin 2\alpha - 2\alpha)}\right]m = 1 \tag{35}$$

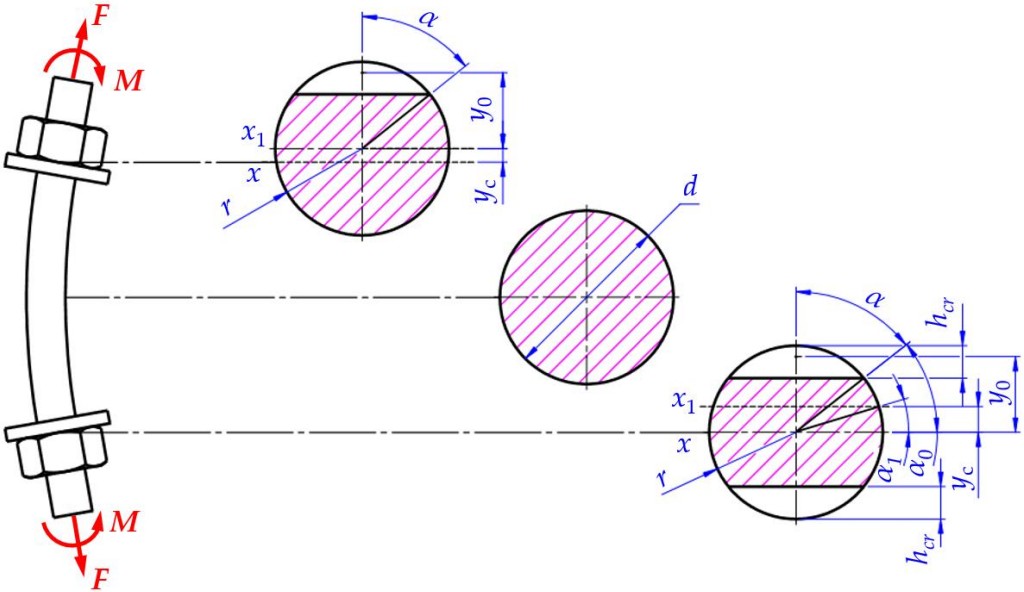

**Figure 16.** Loading of the stud and crack propagation.

The crack depth $h_{cr}$ to the radius ratio of the cross-section $k = h_{cr}/r$. By using those ratios according to Figure 16 and from Equation (35), the conditions of progressive plastic strain were obtained (Table 1).

**Table 1.** Conditions of progressive plastic strain.

| Relative Crack Depth $k = h_{cr}/r$ | Conditions of Progressive Plastic Fracture |
|:---:|:---:|
| $k = 0$ | $n + 0.72m = 1$ |
| $k = 0.05$ | $1.014n + 0.735m = 1$ |
| $k = 0.10$ | $1.04n + 0.768m = 1$ |
| $k = 0.15$ | $1.074n + 0.808m = 1$ |
| $k = 0.20$ | $1.117n + 0.852m = 1$ |

The safety factor was calculated using Equation (27), where the probability ranged from 1% to 99%, changing from 1.073 to 1.791. The developed probabilistic dependencies on the coordinates $P$-$n_d$ and $P$-$m_d$ are presented in Figure 17.

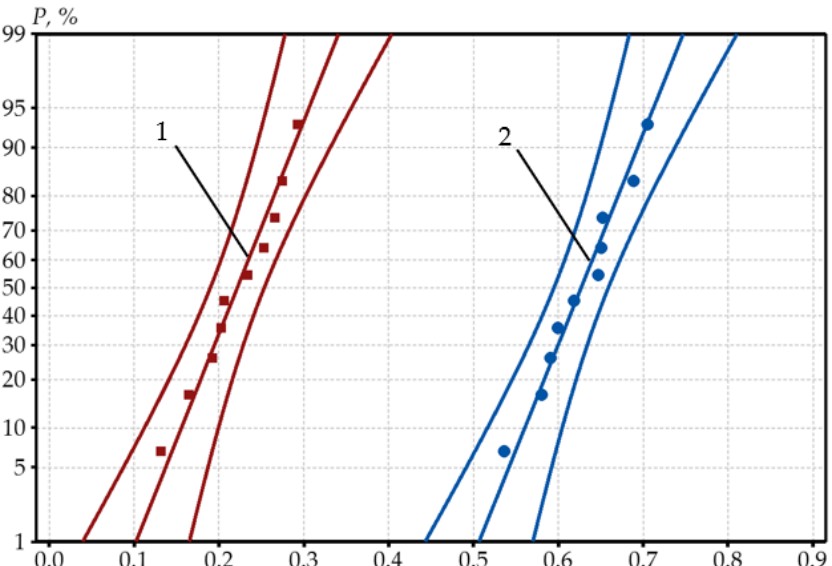

**Figure 17.** Probabilistic relationships of the parameters $m_d$ (1) and $n_d$ (2) and their 95% probabilistic intervals for crack-free threaded joints.

The hypothetical probabilistic curve obtained was used to determine the probabilistic values for the parameters *n* and *m*, where *P* = 1%, 10%, 30%, 50%, 70%, 90%, and 99%. The results obtained are depicted in the shakedown diagram (Figure 18).

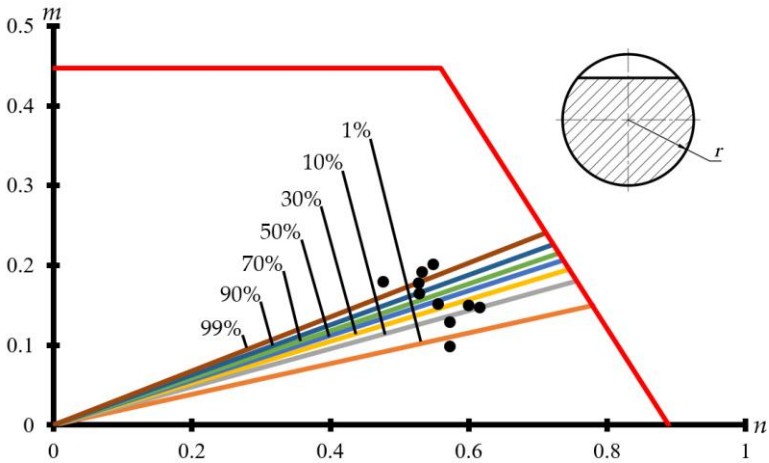

**Figure 18.** Statistical assessment of threaded joints with a one-sided crack: •—experimental data.

### 4.3. Joints with a Two-Sided Crack

For the tightened threaded joint, the position of the nuts may contribute to the development of a symmetric two-sided crack in the cyclic bending plane, as shown in Figure 16. In this case, according to the assumption of ideal elasticity, the variable components of the stresses were calculated using the following equation:

$$\sigma_z^{(e)} = \frac{M}{I_x}y, \quad -r \cdot \cos \alpha_0 \leq y \leq r \cdot \cos \alpha_0 \tag{36}$$

For the fictive yield surface, the stresses were calculated using the following equations:

$$\sigma_{ij}^0 = \begin{cases} \sigma_y - \frac{M}{I_x}y, & 0 \leq y \leq r \cdot \cos \alpha_0 \\ \sigma_y + \frac{M}{I_x}y, & -r \cdot \cos \alpha_0 \leq y \leq 0 \end{cases} \tag{37}$$

The geometric indicators of the two-sided crack cross-section were calculated according to Figure 16. The progressive shape change condition:

$$F \cdot \Delta u_{z0} = 4 \int_0^{l_0} dz \int_0^{r\cos\alpha} dy \int_0^{\sqrt{r^2 - y^2}} \left( \sigma_y - \frac{M}{I_x} \right) \frac{\Delta u_{z0}}{l_0} dx \tag{38}$$

By integration Equation (38) and introduction of dimensionless load parameters $m$, $n$, the following was obtained:

$$\frac{\pi}{\sin 2\alpha + \pi - 2\alpha} n + \frac{16r^4}{9I_x} \left( \frac{1 - \sin^3\alpha}{\pi + \sin 2\alpha - 2\alpha} \right) m = 1 \tag{39}$$

The trigonometric functions of the angle $\alpha$ were expressed by the depth of the crack $h_{cr}$ and the radius $r$. Crack depth $2h_{cr}$ to cross-sectional diameter $d = 2r$ ratio $k = h_{cr}/r$. Using various values of the ratio, progressive plastic strain conditions were obtained (Table 2).

**Table 2.** Conditions of progressive plastic strain.

| Relative Crack Depth $k = h_{cr}/r$ | Conditions of Progressive Plastic Fracture |
|:---:|:---:|
| $k = 0$ | $n + 0.72m = 1$ |
| $k = 0.05$ | $1.013n + 0.745m = 1$ |
| $k = 0.10$ | $1.039n + 0.791m = 1$ |
| $k = 0.15$ | $1.074n + 0.854m = 1$ |
| $k = 0.20$ | $1.116n + 0.932m = 1$ |

The safety factor was calculated using Equation (27), where the probability varied from 1% to 99%, changing from 1.084 to 1.685. The developed probabilistic dependencies on the coordinates $P$-$n_d$ and $P$-$m_d$ are presented in Figure 19.

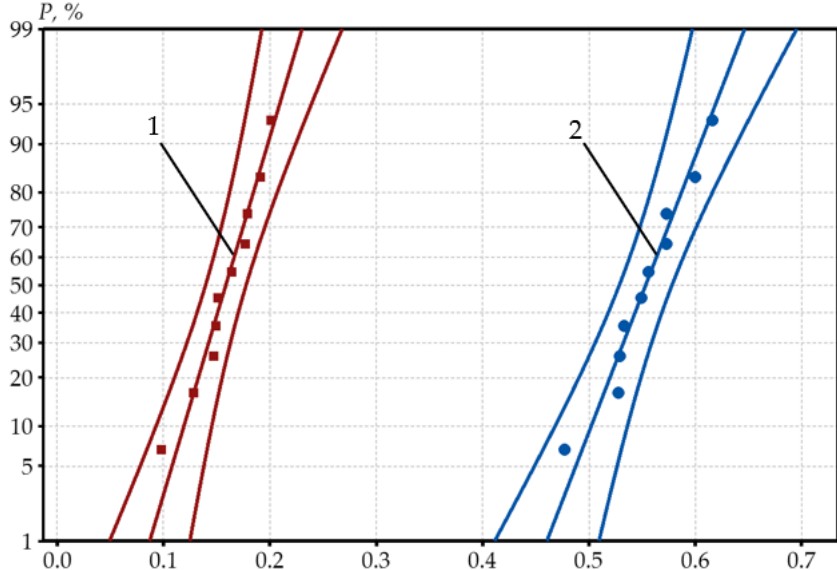

**Figure 19.** Probabilistic relationships of the parameters $m_d$ (1) and $n_d$ (2) and their 95% probabilistic intervals for threaded joints with a two-sided crack.

The hypothetical probabilistic curve obtained was used to determine the probabilistic values for the parameters $n$ and $m$, where $P = 1\%$, $10\%$, $30\%$, $50\%$, $70\%$, $90\%$, and $99\%$. The results obtained are depicted in the shakedown diagram (Figure 20).

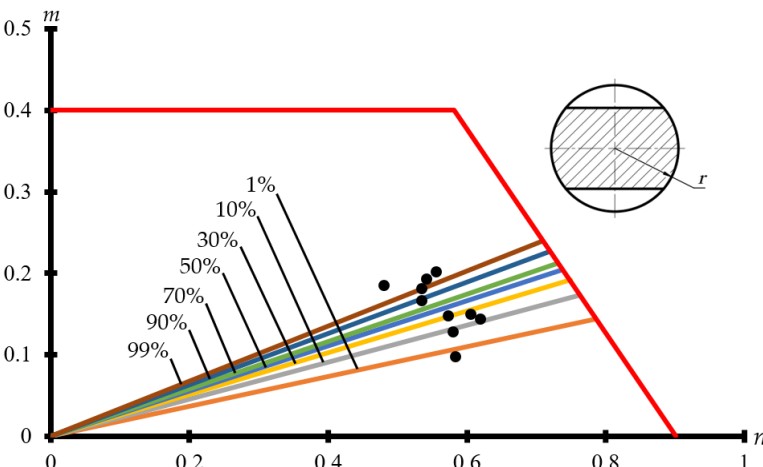

**Figure 20.** Statistical assessment of threaded joints with a two-sided crack: ●—experimental data.

The limit cycle parameters (limit changes of mechanical loads and temperature fields) have maximum values, in which case the shakedown conditions are satisfied in view of the task analyzed, and minimum values, in which case the non-shakedown conditions are satisfied. The reliability of the structure increases when this kind of comprehensive assessment is used.

## 5. Discussion

To evaluate the structural performance of components, it is not sufficient to schematize the actual parts; their overall deformation and the local stress–strain behavior need to be considered, using dependencies that relate the displacements of the model to the external loads due to the occurrence and propagation of a fatigue crack, to the accumulation of plastic deformations, or to the adaptation of the component to cyclical effects, which are influenced by the interaction between the components.

The experimental and theoretical investigation of the resistance of threaded joints to low-cycle and multicycle loading shows that the existing calculation methodologies are not sufficiently valid, and therefore the existing methods of investigation have now been refined and improved, taking into account structural, technological, and operational factors, using decay mechanics criteria and theories of shakedown.

The analytical dependences of plastic failure and progressive shape change for studs without a crack, taking into account parameters of the failure process obtained in this work such as crack shape, depth, and stress reduction have been validated experimentally. The plastic degradation limit state is unrealistic in threaded joints used to join assemblies and flanges.

The analytical shakedown conditions are accurate as they are obtained using static and kinematic methods, i.e., the result is approached from two sides. The condition of progressive change of shape can be used as a limit state, which occurs in the boundary layers of the stud if the sum of the tensile and bending stresses is close to 1,2. When this condition is exceeded, unidirectional accumulation of plastic deformation begins.

Using the analytical expressions obtained and the shakedown diagrams derived therefrom, the paper presents a new methodology for the calculation of the margin of progressive change in shape for threaded joints without a crack and with one-way and two-way cracks.

A completely new methodology for the calculation of the progressive change in shape is suggested to be performed prior to the calculation of the cyclic strength to form, together with the resulting or refined dependencies, a unified methodology for the calculation of the cyclic strength and the shakedown of the responsible threaded joint.

In the paper the shakedown analysis of the threaded joints was performed using the static and kinematic methods, i.e., the result was approached using a two-sided tech-

nique. The shakedown conditions were obtained for crack-free studs in view of the experimental data, i.e., the failure process parameters such as the shape, size of crack, and stress reduction.

Tightening and cyclic bending of the threaded joints may cause accumulation of plastic strain in the outer layers of the stud. In case a favorable residual stress field has developed, the cyclic plastic strain does not accumulate, i.e., the shakedown takes place.

The limit state conditions determined for crack-free studs and studs with one-sided and two-sided cracks show that studs with two-sided cracks have the smallest safety margin for progressive shape change.

Calculation of the cyclic strength and shakedown is an important stage in the design of the critical threaded joints and connecting elements and crucial for service life extension. The calculation methods designed and improved within the study were based on theoretical and experimental investigations and simplified for convenient application to engineering practice.

## 6. Conclusions

Based on the results, we make the following observations:

(1) Short cracks (up to 0.5 mm deep, up to 5 mm long) do not reduce the static strength of threaded connections. This is the size of the crack easily measurable by nondestructive inspection methods, which can be considered as the maximum allowed under service conditions.

(2) The crack propagation characteristics are different in cyclically bending and tensile threaded joints. At a certain fixed nut-and-pin position (the bending plane is aligned with the most loaded position in the thread recess), the crack is one-sided. Otherwise, i.e., at any other position of the nut relative to the stud, the crack is two-sided. The adaptability of the crack decreases with the progression of a two-sided crack, i.e., the crack is more dangerous.

(3) In prestressed and cyclically bending threaded joints, a relationship was found between the loosening and the parameters of the failure process, such as the shape and depth of the crack. The propagation of the crack is mainly due to the energy of the cyclic bending, so the stresses in the propagation plane of the crack increase as the stress is only slightly reduced. When the crack reaches a critical size, the rest of the crack breaks rapidly due to stress after a small pulse of cyclic bending energy.

(4) In order to obtain a stable tension of the threaded joint and thus a stable clamping of the elements to be joined, it is necessary to apply a repeated (3–4 times) turn of the nut, irrespective of the method used to assemble the threaded joint. This prevents stress reduction during the assembly process.

(5) Shakedown analysis of the threaded joints was performed. The safety margin of the progressive shape change in the experimentally tested joints was determined, and a statistical evaluation was performed. The safety factor ranged from 1.06 to 1.79 for the joints analyzed, with the probability varying from 1% to 99%.

(6) Where the tightening was close to $0.8\sigma_y$, ($\sigma_y$—yield strength) and the bending was $0.4\sigma_y$, the safety factor for the progressive change in shape was close to 1. These safety factors are considered to have insufficient reliability in the elastic–plastic area analyzed.

(7) The statistical investigation carried out as part of the study showed that to develop a reliable definition of the safety margin for progressive change in shape in threaded joints, an experimental curve of 50% must be developed, which, in the case analyzed, reflects a safety factor equal to 1.322.

**Author Contributions:** Conceptualization, Ž.B. and M.L.; methodology, Ž.B. and M.L.; software, V.L.; validation, Ž.B., M.L. and V.L.; formal analysis, Ž.B., M.L. and V.L.; investigation, Ž.B., M.L, V.L. and L.R.; resources, L.R.; data curation, Ž.B., M.L, V.L. and L.R.; writing—original draft preparation, Ž.B., M.L. and V.L.; writing—review and editing, Ž.B., M.L. and V.L.; visualization, V.L.; supervision, Ž.B., M.L., V.L. and L.R; project administration, L.R. All authors have read and agreed to the published version of the manuscript.

**Funding:** This research received no external funding.

**Institutional Review Board Statement:** Not applicable.

**Informed Consent Statement:** This article does not contain any studies with human participants or animals performed by the author.

**Data Availability Statement:** No new data were created or analyzed in this study. Data sharing is not applicable to this article.

**Conflicts of Interest:** The authors declare no conflict of interest.

## Nomenclature

| | |
|---|---|
| $A$ | cross-section area of the stud (mm$^2$); |
| $A_e$, $C$, $S_e$ | parameters depending on the mechanical properties of steel and indicators of the load cycle ($A_e = 0.149$; $C = 0.5$; $S_e = 0.45$); |
| $A_p$ | area of fictive yield surface (mm$^2$); |
| $a$, $b$, $m$, $n$ | dimensionless load parameters; |
| $D$ | plastic energy dissipation; |
| $d$ | diameter of the stud (mm); |
| $E$ | elasticity modulus (MPa); |
| $e_c$ | material plasticity indicator determined by assessment of the variation in the cross-section area of the standard cylindrical specimen subjected to tension; |
| $F$, $F_t$ | axial and tensile force (N); |
| $F_y$ | limiting axial force causing plastic deformation (N); |
| $F_{fd}$ | reaction force in the clamped parts; |
| $F\left(\varepsilon_{ij0}''\right)$ | plastic energy dissipative function; |
| $h_a$, $h_{cr}$ | crack depth (mm); |
| $I_x$, $I_{x1}$ | moments of inertia of the cross-section (mm$^4$); |
| $l_0$ | effective length of the stud (distance between the cross-sections subjected to the largest loading in the nut–stud–nut assembly, mm); |
| $M$ | variable bending moment (N * mm); |
| $M_f$, $M^*$ | threshold bending moment (N * mm); |
| $M_y$ | limiting bending moment causing plastic deformation (N*mm); |
| $m_0$ | number of specimens in the selected interval; |
| $N$ | number of cycles; |
| $N_{adm}$ | admissible number of cycles; |
| $N_f$ | number of cycles to failure; |
| $n_0$ | number of tested specimens; |
| $n_N$ | safety factor of strength by the number of cycles; |
| $n_\sigma$ | safety factor of strength by stress; |
| $P$ | random value probability; |
| $P(x)$ | probability density function; |
| $r$ | stud radius (mm); |
| $r_a$ | cycle asymmetry coefficient; |
| $r_c$ | thread root radius (mm); |
| $T$, $t$ | time (s); |
| $u_{i0}$ | plastic displacement components (mm); |
| $\Delta u_{i0}$ | plastic displacement increment components (mm); |
| $\Delta u_{z0}$ | elongation of the stud (mm); |
| $V$ | volume (mm$^3$); |
| $x_i^0$, $p_i^0$, $p_z^0$ | volumetric and superficial force components; |

| | |
|---|---|
| $x_i$ | random measure; |
| $x_0$ | mean arithmetic value; |
| $x_1$, $y_1$ | coordinates of the neutral axis (mm); |
| $y_0$ | coordinate of the crack (mm); |
| Greek symbols | |
| $\alpha_0$ | angle to crack (°); |
| $\alpha_1$ | angle to neutral axis (°); |
| $\varepsilon''_{ij0}$ | plastic strain components corresponding to $u_{ij0}$; |
| $\Delta\varepsilon''_{ij0}$, $\Delta\varepsilon''_{z0}$ | plastic strain increment components corresponding to $\Delta u_{ij0}$; |
| $\varepsilon$ | stud turns strain (%); |
| $\varepsilon_f$ | failure strain (%); |
| $\varepsilon_d$ | clamped parts strain (%); |
| $\eta$ | safety factor for the progressive shape change; |
| $\sigma_0$ | mean square deviation; |
| $\sigma_{0.2}$ | elastic limit or yield strength (MPa), the stress at which 0.2% plastic strain occurs; |
| $\sigma_{-1}$ | endurance limit of the material (MPa); |
| $^*\sigma_{a,adm}$ | amplitude of the relative local tensile stress cycle (MPa); |
| $\sigma_b$, $\sigma_{b1}$, $\sigma_{b2}$, $\sigma'_b$, $\sigma'_{b1}$, $\sigma'_{b2}$ | bending stress (MPa); |
| $\sigma_{ij}$ | stress of the fictive yield surface (MPa); |
| $\sigma^0_{ij}$ | minimum stress of the fictive yield surface (MPa); |
| $\sigma^0_{ij*}$ | stress of the fictive yield surface related to the $\Delta\varepsilon''_{ij0}$ associated flow rule (MPa); |
| $\sigma^{(e)}_{ij}$, $\sigma^{(e)}_z$ | stress components caused by external loads on a perfectly elastic material (MPa); |
| $\sigma^0_z$ | variable stress component (MPa); |
| $\sigma_f$ | failure stress (MPa); |
| $\sigma_{max}$, $\sigma_{min}$ | maximum and minimum cycle stress (MPa); |
| $\sigma_y$ | yield stress (MPa); |
| $\sigma_S$ | total bending and tensile stress (MPa); |
| $\sigma_t$, $\sigma_{tmax}$ | tension stress (MPa); |
| $\sigma_{td}$ | tension stress reduction due to contact between intermediate parts (MPa); |
| $\sigma_{tb}$ | tension stress reduction due to shear, thread surface contact and crack propagation (MPa); |
| $\sigma_{tn}$ | tension stress reduction due to shear of the first turns of the nut (MPa); |
| $\sigma_t^{\circlearrowright}$ | tension stress reduction due to self-rotation of the nut (MPa); |
| $\sigma_u$ | ultimate tensile stress (MPa); |
| $\psi$ | percent area reduction (%). |
| Abbreviations | |
| PNAE norms | Regularities and Norms in Nuclear Power Engineering; |
| ASME norms | The American Society of Mechanical Engineers. Boiler and Pressure Vessel Code, an internationally recognized code. |

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
