# Peer review of "Assessment of the Durability of Threaded Joints"

_applsci, doi:10.3390/app112412162_

Round 1
Reviewer 1 Report
An interesting paper where a lot of experimental data have been collected regarding durability of threaded joints under cyclic loading. Based on these data a kinetic fatigue diagram was developed according to failure mechanics criteria (covers the areas of low-cycle and multicycle failure) and a statistical evaluation of the failure was performed. The shakedown conditions in the threaded joints subjected to tension and bending were also determined. The article is very well structured and authors presented clearly the aims of the paper and the reasons for the investigation.
The paper content and presented results are very useful and valuable from practical point of view, particularly for the design engineers dealing with nuclear power installations. In addition, the article provide some scientific contribution. The methodology and equipment employed by the authors to reach the paper objectives is combination of standard and original (self-designed) elements. English language is correct with a few minor grammatical and typing errors.
In general, this article is a solid work with nothing much to review. But, before publishing I suggest the authors to provide some revisions and explanations according the list below in order to increase the paper quality.
- What type of tension sensor(s) was/were built-in bending equipment (Fig.2)? Are there any differences in the sensors marked with numbers 6 and 7? It will be useful to add in manuscript a picture/photo of the assembly – testing machine + fixture + measuring equipment.
- Details regarding the type of thread (rolled or cut thread) on the stud and surfaces quality are missing. Namely, the surface quality (friction) and the manufacturing methods have a significant influence on (exploitation) characteristics and failure of treaded joins.
- Bolts (studs) and nuts connecting components of pressure vessels and piping elements used in power plants are usually exposed to high temperatures (thermal loading). Considering that your tests were performed at room temperature how useful/reliable are obtained results in term of improvement of calculation norms for power plant and piping installations?
- At a certain fixed nut-and-pin position (the bending plane is aligned with the most loaded position in the thread recess), the crack is one-sided. Otherwise, i.e., at any other position of the nut relative to the stud, the gap is two-sided. Please explain more detailed.
Reviewer 2 Report
Re: Assessment of the durability of threaded joints
Z.Bazaras et al.
The paper is devoted to analysis of fatigue failure of threaded joints. The topic is of actual engineering and research interest, still requiring further study. The presented analysis is combined of experimental and analytical portions, referring to crack initiation and propagation, thread loosening effects, plastic deformation and shake-down analysis. Regrettably, the text is not clearly written and requires essential improvements.
Section 2 is aimed to formulate main study objectives, but they are not clearly exposed in relation to existing methodology. Some corrections are needed.
Line 140: put “reaches” instead of “teaches”
Line 162: put ra=σ min/σmax=0.6 - 0.7
Line 171: put “plastic state” instead of “plastic area”
Line 175: put “plastic strain of varying sign”
Section 3 presents the main experimental results of the paper. It should have the title: “Experimental results and their analysis”. In presenting parameters of an used in tests steel, only yield- and failure stress values, plus undefined parameter ψ are listed (line 182), but the elastic moduli are not included. In describing the experimental procedure for combined bending-tension testing of the threaded joint, the imposed control of strain or stress, curvature or moment, should be clearly defined and graphically illustrated. The text does not provide clear control characterization. The reviewer understands that a mixed control is applied, by first the tightening process in axial tension at assumed increasing strain value and next by the bending moment control for cyclic loading at fixed axial load, cf. Figs. 3, 4, 5.
The reference to PNA norm or ASME code, Eqs. 1 and 2, was made comparing own experimental results with code fatigue diagrams. It is believed the these formulae are incorrectly presented. As rc is the thread root radius, it should not occur in these expressions , only ra presenting cycle non-symmetry. The brackets are not correctly placed in the second formula.
The analysis of crack propagation is limited to description of one-and two-sided cracks, but no specific results are presented related to initiation and propagation stages.
The analysis of thread loosening effect is important and lacks consistent characterization. The evolution of tensile stress was determined experimentally and shown in Fig.11. The presented schematically loading diagrams, Figs. 9 and 10, illustrate schematically the effects of tightening, plastic strain and cracking, but there are no calculation formulae providing numerical assessment of loosening effect and parameters needed for its determination.
Line 413: non understandable sentence.
Section 4 is numerated in the paper as Section 5. It is related to assessment of axial elongation induced by flexural cycles superposed on axially pre-tensioned studs. The kinematic shake-down theorem of Koiter was applied to specify upper bound on the shake-down load. However, the analysis presented and Eqs. (5)-(15) are obscure and contain errors, cf. Eqs. (9) and (10). This section could be an important part of paper, but all formulae should be clearly presented . Some details of calculation could be put in appendix.
To be qualified for publication, the paper needs essential modifications.
Reviewer 3 Report
My "Comments and Suggestions for Authors
" are presented in the attached PDF.

Round 2
Reviewer 2 Report
The text has been slightly improved, but Section 4 on shake down analysis is still obscure. The upper bound Eq. (4) has improper inequality sign, also used symbols are undefined. In Eq. (7) the expression for energy dissipation seems incorrect. For beams of rectangular and circular cross sections under constant axial stress and oscillating bending moment the domains of elastic, shake-down and ratcheting response are analytically expressed (cf. book : J. A. Konig: Shake down for elastic-plastic structures, Elsevier-PWN, 1987). Fig. 12 should present more accurately these domains. Eqs. (5)-12) should be assisted by cross section geometry diagram.
Reviewer 3 Report
After analysing again the article as a whole, I keep my opinion. The manuscript looks more like a technical report (or chapter of a book) than a scientific article.
In my previous comments, I have asked: - What are the new findings of the present manuscript? The question “Can the authors write the “highlights” of the present manuscript?” was just to help the authors to improve the manuscript; not to include those “highlights” in the paper.
If this manuscript was submitted to a journal specially dedicated to fatigue & fracture, certainly it would receive further criticism.
